# Investigation of Metabolic Resistance to Soybean Aphid (*Aphis glycines* Matsumura) Feeding in Soybean Cultivars

**DOI:** 10.3390/insects13040356

**Published:** 2022-04-05

**Authors:** Ian M. Scott, Tim McDowell, Justin B. Renaud, Sophie W. Krolikowski, Ling Chen, Sangeeta Dhaubhadel

**Affiliations:** Agriculture and Agri-Food Canada, London Research and Development Centre, London, ON N5V 4T3, Canada; tim.mcdowell@agr.gc.ca (T.M.); justin.renaud@agr.gc.ca (J.B.R.); sophie.krolikowski@agr.gc.ca (S.W.K.); ling.chen@agr.gc.ca (L.C.); sangeeta.dhaubhadel@agr.gc.ca (S.D.)

**Keywords:** *Glycine max*, *Aphis glycines*, isoflavonoids, free amino acids, host plant resistance

## Abstract

**Simple Summary:**

This project examined the interaction between soybean aphids and Ontario-grown soybean cultivars to determine which leaf metabolites were most associated with aphid resistance. Tolerance and resistance were determined by measuring the growth and reproduction of aphids and leaf feeding damage over 10-day and 4-week infestation periods. Chromatographic techniques were used for the analysis of legume-specific plant natural products and isoflavonoids, and high-resolution mass spectrometry was used for the identification of free amino acids in aphid-resistant, tolerant, and susceptible soybean cultivars. There was a low correlation between isoflavonoid leaf concentrations and aphid resistance in the soybean varieties studied; however, the aphid-resistant cultivars were determined to have lower free amino acid concentrations, indicating that lower nutrient quality could be responsible for the resistance observed. Identifying these cultivars is important for managing aphid populations and provides an additional tool for soybean integrated pest management.

**Abstract:**

Soybean aphid (*Aphis glycines*) is a major soybean (*Glycine max*) herbivore pest in many soybean growing regions. High numbers of aphids on soybean can cause severe reductions in yield. The management of soybean aphids includes monitoring, insecticide applications when required, and the use of resistant cultivars. Soybean aphid-resistant soybean varieties are associated with genes that confer one or more categories of resistance to soybean aphids, including antibiosis (affects survival, growth, and fecundity), antixenosis (affects behaviour such as feeding), and tolerance (plant can withstand greater damage without economic loss). The genetic resistance of soybean to several herbivores has been associated with isoflavonoid phytoalexins; however, this correlation has not been observed in soybean varieties commonly grown in southern Ontario, Canada. Isoflavonoids in the leaves of 18 cultivars in the early growth stage were analyzed by HPLC and the concentration by fresh weight was used to rate the potential resistance to aphids. Greenhouse and growth cabinet trials determined that the cultivars with greater resistance to aphids were Harosoy 63 and OAC Avatar. The most susceptible cultivar was Maple Arrow, whereas Pagoda and Conrad were more tolerant to aphid feeding damage. Overall, there was a low correlation between the number of aphids per leaf, feeding damage, and leaf isoflavonoid levels. Metabolite profiling by high-resolution LC-MS determined that the most resistant cultivar had on average lower levels of certain free amino acids (Met, Tyr, and His) relative to the most susceptible cultivar. This suggests that within the tested cultivars, nutritional quality stimulates aphid feeding more than isoflavonoids negatively affect aphid feeding or growth. These findings provide a better understanding of soybean host plant resistance and suggest ways to improve soybean resistance to aphid feeding through the breeding or metabolic engineering of leaf metabolites.

## 1. Introduction

Soybean (*Glycine max*. L Merr) is an important field crop in southern Canada, and has been grown in Ontario for over 70 years. In 2019, the amount of soybean acreage planted was greater than 3.1 M acres in Canada with over half (1.6 M acres) in southwestern Ontario [1], where yields of 45 to 52 bushels per acre were reported and 1.7 to 2.1 M tonnes were produced. Most of the soybean produced is glyphosate tolerant (75%) and used for oil and animal feed, while the remainder is non-GMO and is grown for food and organic production [2]. In southwestern Ontario, the maturity groups range from MG I to MG III, and the cultivars are selected by their performance in field trials that also consider disease and pest resistance. One of the most damaging insect pests of Ontario soybean is the soybean aphid (*Aphis glycines* Matsumura; Hempitera: Aphididae), an invasive species that became established in 2000. The yield loss caused by soybean aphid is greatest in the R1–R2 stages, when flowers can abort and impact pod establishment. Beyond the R3 stage, damage can result in a smaller seed size and a reduction in seed quality [2]. The current strategy for *A. glycines* management involves monitoring and reacting by applying insecticides to reduce the aphid pressure. Insecticides are applied to control high populations of aphids; however, the over-use of insecticides has a negative impact on native enemies, such as ladybird beetles. Another integrated pest management (IPM) strategy is the use of host plant resistance, which is a heritable decrease in plant susceptibility to the pest [3]. Reducing the number of insecticide sprays or delaying applications until later in the growing season by slowing aphid population growth is the goal of using more resistant soybean cultivars.

Many studies have screened existing soybean varieties for antibiosis, antixenosis, and tolerance to soybean aphids [3,4,5,6,7,8]; however, they generally have not associated the resistance with mechanisms such as physical or chemical defences. Twelve genes associated with resistance to *A. glycines* (*Rag*) have been identified that affect the biology (antibiosis) and behaviour (antixenosis) of soybean aphids [9]. One of the most comprehensive studies integrated insect resistance phenotypes from the Germplasm Resources Information Network (GRIN) and genotypes (SNPs) from the Soybean Germplasm Collection (USDA-ARS) for six soybean insects [10]. A recent review on the soybean—*A. glycines* interaction discussed the need for further studies to examine the role of chemical defences for aphid-resistant soybean genotypes by quantifying defence-related metabolite levels [11]. They encouraged research on soybean chemical defences to identify the specific metabolites employed in the resistant response and their correlation with changes in aphid feeding behaviour.

Tolerance is the ability of plants to withstand or recover from insect damage, antibiosis resistance directly affects the insect and causes reduced abundance, and antixenosis resistance affects insect behaviour and may reduce the preference for the plant. The development of a soybean genotype that is tolerant to *A. glycines* is thought to benefit IPM programs by increasing the economic injury level, allowing for the adoption of a higher economic threshold, reducing the number of insecticide applications, and decreasing the selection pressure for resistant biotypes to antibiotic or antixenotic plants [4]. Another aspect of host plant resistance that should be examined is how the resistant lines would fare with other aphid biotypes known to be present in southern Ontario [12].

Host plant resistance is generally cultivar-specific and is defined by chemical and physical factors, or a combination of both. An important phytochemical defence in legumes is isoflavonoid phytoalexins. Pathogens induce soybean leaf levels of isoflavonoids (daidzein, formononetin, isoformononetin, genistein, glycitein, and glyceollins I–III) and isoflavone glycosides (daidzin, genistin, ononin) and their malonylglycoside conjugates (malonyldaidzin and malonylgenistin) with varying concentrations and accumulation between cultivars [13]. Genistein, daidzein, glycitein, and their conjugates are found in high amounts in leaf and embryo tissues during late-stage seed development, while soybean seeds contain more malonyl glycosides, all of which act to modulate microbe interactions [14]. Isoflavonoids can also provide host plant resistance to insects, for example, aphids and stink bugs [15,16]. Since both stink bugs and aphids are stylet feeders, they would encounter isoflavones as soluble and mobile glycosylated derivatives present in the soybean phloem [14], but more likely in the parenchyma and epidermal tissues [17]. One study showed that the feeding behaviour of the pea aphid (*Acyrthosiphon pisum*) was prolonged by genistein, as demonstrated by a longer period of stylet probing, reduced salivation, and passive ingestion [18]. In another study the soybean genotype ‘Zhongdou 27’ from China with more than twice the amount of daidzein, 80% more genistein, 50% more glycitein, and almost double the total isoflavones was shown to be more resistant to soybean aphid [16]. It was also observed that aphid attacks increased the levels of daidzein and genistein in the resistant but not the susceptible genotypes. Screening the amounts of phytoalexins in resistant cultivars is promoted as a way to identify insect-resistant germplasm for breeding programmes. For example, one multivariate analysis indicated that daidzein and genistein were the most important discriminators that reduced damage from the seed pod-feeding brown stink bug (*Euschistus heros*) in soybean [19].

A parallel study examined host plant resistance in soybean to the two-spotted spider mite *Tetranychus urticae* (Koch; Acari: Tetranychidae) and isoflavonoid levels were found to have a low correlation with resistance [20]. The more relevant leaf chemistry that was associated with spider mite-resistant and susceptible soybean cultivars was the presence of free amino acids and other metabolites. In this concurrent study, the research objectives were to (1) investigate genetic resistance to soybean aphid in the same soybean cultivars and (2) analyse the concentration of isoflavonoids and other metabolites in the leaves of susceptible and resistant cultivars in order to develop recommendations for cultivars that could be the focus of field trials for potential use in breeding further resistance.

## 2. Materials and Methods

### 2.1. Plants

Soybean seeds were obtained from germplasm banks (Agriculture and Agri-Food Canada (AAFC), London Research and Development Centre) and from Dr. I. Rajcan and Dr. M. Eskandari (University of Guelph, Guelph, ON, Canada). Newly developed soybean aphid-resistant lines (E16 265, E16 266, and E16 267) were also obtained through the breeders at the University of Guelph. The seeds were planted in pots with Pro-Mix BX Mycorrhizae^TM^ soil (Premier Tech Home and Garden, Rivière-du-Loup, QC, Canada) and grown in a growth chamber set at 23 °C under a light intensity of 250 µmol photons m^−2^s^−1^ for 16:8 h light:dark (L:D) cycle with 60–70% relative humidity (RH). Plants were fertilized with N:P:K 20:8:20 and watered with reversed osmosis (RO) water when required.

### 2.2. Aphids

Soybean aphids (biotype 1) were obtained from Dr. J. Brodeur (University of Montreal) and maintained on the susceptible isoflavonoid soybean cultivar Beer Friend. Separate mesh cages (50 cm^3^) held two soybean plants and were infested with aphids; plants were replaced as required. The cages with plants and aphids were kept in a growth chamber at 26 ± 2 °C and 50 ± 5% RH for a 16:8 L:D photoperiod. A fresh 3–4-week-old plant was added to each colony mesh cage every 1–2 weeks after removing the oldest plant.

### 2.3. Soybean Aphid Resistance Assessment

#### 2.3.1. Ten-Day Trial—Survival and Fecundity on a Soybean Leaf

Soybean plants at the V3–V4 stage—when third or fourth trifoliate leaves had developed—were grown under the conditions described previously for use in a 10-day bioassay with aphids to determine the differences in nymph growth and the fecundity of adults that developed within one generation. Soybean cultivars with representative low, mid, and high isoflavonoid leaf levels based on the average concentration of measurements taken at the V1 and V3 stages were selected for the trials [20]. An additional 3 aphid-resistant lines were added to this test as positive control lines. Five newly emerged aphid nymphs (<1 day old) were placed on the mid leaf of both the 2nd and 3rd true leaves of the soybean plant in the V3–V4 stage for 10 days. Lanolin was spread around the petiole to isolate the aphids to the leaf and a paper clip, wooden stake, and pipe cleaners held back the adjacent leaves from the 2 isolated leaves. At the end of the 10th day, the number of nymph and adult aphids on each leaf were counted to assess aphid population growth. Four replicate plants per cultivar were used in each trial for the aphids. For each cultivar, one plant without aphids and one plant with the lanolin, paper clips, stake, and pipe cleaners but no aphids were used as controls to determine the range of leaf isoflavonoids for comparison to the aphid and leaf isolation set-up plants. The plants were held in a growth cabinet or environmental room with conditions set as follows: 26 ± 2 °C, 60 ± 5% RH, 250 µmol photons m^−2^s^−1^ light intensity, and a 16:8 h L:D light period.

#### 2.3.2. Four-Week Trial—Sample Counts and Leaf Damage Assessment

Soybean plants at the V1 stage grown under the conditions described previously were used in a four-week trial with aphids to determine the level of infestation by sampling the population size and feeding damage on the top leaves of the plants on week 4. Plants from soybean cultivars with varying leaf isoflavonoid levels at the V1 and V3 stages were selected for the trials. Three greenhouse trials using 4 cultivars per trial were conducted for a 4-week period based on the methods used in a previous study [5]. At the beginning of the trial, each V1 plant was enclosed in a white mesh cage (50 cm^3^). The soybean plants were inoculated with two wingless adult aphids, randomly selected from the colony, that were placed on the first full mid leaf before being enclosed in the cage. The control plants received no aphids. All the plant cages with control and aphid treatments were transferred to the greenhouse where the environmental conditions were maintained at 24 ± 2 °C, 50 ± 2% RH, 250 µmol light, 16:8 h L:D. Each week (weeks 1–3), the cages were monitored for plant growth, aphid damage, and soil moisture and were watered as required. The plants and soil were sprayed lightly with a fertilizer solution (20–8–20, 0.5 g/L water) each week. After 4 weeks (week 4), the leaves were removed from the plant and photographed (top and bottom sides), and then scanned (top side only) with an Expression 11000XL photo scanner (Epson, Markham, ON, Canada). All leaves from each of the 3 trifoliate leaf levels were combined in 3 separate 15 mL Falcon tubes, flash frozen in liquid nitrogen, and transferred to storage in the −80 °C freezer. The aphid infestation was visually rated for each plant by counting the numbers per leaf on the top 3 trifoliate leaves (top and bottom sides) and completing a damage assessment using percent leaf area damage. Antibiosis was assessed based on the average count of aphids (nymphs and adults) per trifoliate leaves for each cultivar. Tolerance was assessed using the percent leaf area damage by comparing the colour of the control plant leaves to the aphid-infested leaves (WinFOLIA Pro, Regent Instruments, Nepean, ON, Canada). For each cultivar, new assessment parameters were established in WinFOLIA for the colour of the healthy control leaves first, followed by the colour of aphid-damaged areas on the infested leaves. The program analyzed the amount of healthy leaf colour on the aphid-damaged leaf, which was then converted to percent leaf damage based on the difference.

### 2.4. Isoflavonoid Analyses

Leaf tissues were ground and extracted in 1:1 (*v*/*v*) acetonitrile:water following the method described in our previous spider mite [20] and soybean isoflavonoid studies [14,21]. Briefly, malonyl- and acetyl-isoflavonoid conjugates were converted to their corresponding glycosides by hydrolysing 100 µL of leaf extracts with 50 µL 5% KOH solution. The samples were incubated at room temperature for 5 h, followed by neutralization with 50 µL 14% KH_2_PO_4_. The analyses were completed using our previously described methods [20] with an Agilent 1200 series HPLC-ELSD and a C18 column (Symmetry^®^, 4.6 × 250 mm). The mobile phase gradient was 10–35% acetonitrile in 0.1% acetic acid over 45 min at a flow rate of 1 mL min^−1^ and the detector was set at 260 nm. The concentrations of aglycones (daidzein, glyceitin, and genistein) and their glycosides (daidzin, glycitin, and genistin) were identified and quantified by the comparison of retention time and UV spectra to their corresponding standards. The standards of each isoflavonoid were purchased from Sigma-Aldrich (Oakville, ON, Canada).

### 2.5. Metabolomic Analyses

The soybean leaf metabolomics analysis followed the methods from our previous study [20]. Extraction was performed in methanol:water (4:1 *v*/*v*) with sonication for 15 min followed by centrifugation at 11,000× *g* for 10 min at 4 °C. The samples were analyzed using a Thermo Scientific ™ Q-Exactive™ Orbitrap Mass Spectrometer (Thermo Fisher Scientific, Mississauga, ON, Canada) coupled to an Agilent 1290 HPLC. The following conditions were used for heated electrospray ionization (HESI): capillary voltage, 5.0 kV; capillary temperature, 330 °C; sheath gas, 32 arbitrary units; auxiliary gas, 10 units; probe heater temperature, 280 °C; and S-lens RF level, 50%. The analytes were resolved by hydrophilic liquid interaction chromatography (HILIC) with a 350 µL min^−1^ flow rate. Samples (3 µL) were injected onto an Agilent HILIC-Z (2.1 × 100 mm, 2.7 µm; Agilent, Mississauga, ON, Canada) column maintained at 35 °C. Compounds were resolved with mobile phases of 20 mM ammonium formate in water (A) and 20 mM ammonium formate in 90% acetonitrile (B) operating with the following gradient: 0 min, 100% B; 0.5 min, 100% B; 5.3 min, 80% B; 9.5 min, 30% B; 13.5 min, 30% B, 14.5 min 100% B; and 15.5 min, 90% B. The samples were analyzed in both positive and negative ionization modes, at a resolution of 70,000, automatic gain control (AGC) of 5 × 10^5^, maximum injection time (IT) of 256 ms, and mass range of 70 to 950 *m*/*z*. Composite samples were also analyzed by a top 3 data-dependent acquisition for compound identification with a 17,500 resolution, 1 × 10^5^ AGC, 64 ms, 1.2 *m*/*z* isolation window, and 28 normalized collision energy.

Thermo raw files were converted to the mzml format using Protewizard with the peak filter applied [22]. Features were detected using the XCMS package [23] with the centWave method (ppm tolerance 1.0) [24]. The signal to noise threshold was set to 5, noise was set to 5 × 10^6^, and pre-filter was set to six scans with a minimum intensity of 5000. Retention time correction was conducted using the obiwarp method [25]. The grouping of features was set to those present in at least 25% of all samples (retention time deviation 10 s; *m*/*z* width, 0.015). The ‘fillPeaks’ function was applied with default settings. Remaining zero values were input with two-thirds of the minimum value on a per mass basis. Compounds were identified by the comparison of retention times and *m*/*z* to authentic standards or by accurate mass and the comparison of fragmentation patterns to MS/MS databases. Amino acid concentrations in each sample were estimated by external calibration (TraceCert1Amino Acids Mix Solution, Millipore Sigma, St. Louis, MO, USA) under the assumption that signal suppression/enhancement would be approximately equal across all soybean leaf samples. These data have been submitted to the NIH Common Fund’s National Metabolomics Data Repository (NMDR) website, the Metabolomics Workbench, https://www.metabolomicsworkbench.org (accessed on 17 February 2022).

The Benjamini-Hochberg multiple hypothesis correction was applied to determine molecular features that were significantly different in the compared cultivars. 

### 2.6. Statistical Analyses

A two-way analysis of variance (ANOVA) was used to assess significant effects between the various cultivars and trifoliate leaves regarding their isoflavonoid concentration and the aphid number and fecundity parameters. A Tukey’s test was applied to measure post hoc comparisons. Data were transformed when necessary; however, if the transformed data failed normality and homogeneity tests, the analysis was completed with a two-way (leaf damage in 4-week trial) or three-way ANOVA (isoflavonoid concentrations in the 10-day trial) using type-two error statements. In the case of the aphid counts on the 3rd leaf of soybean plants in the 10-day trial, a non-parametric Kruskal-Wallis test was used with comparisons performed using Dunn’s test. The number of aphid nymphs and adults were correlated with the isoflavonoid concentration of each soybean cultivar at the growth stages associated with the sampling date using Pearson’s correlation log rank analyses. Metabolite data from 2 resistant, 2 tolerant, and 1 susceptible cultivar were analyzed with two-way ANOVA using type-two error statements. All statistical analyses were performed with R software, v3.2.5, and the RStudio tool [26].

## 3. Results

### 3.1. Relative Aphid Resistance Based on Isoflavonoid Levels in Soybean Cultivars—Ten-Day Trial

In a previous study, to rank the selected Ontario-grown soybean cultivars based on leaf isoflavonoid levels, we analysed the leaf extracts from 18 different soybean cultivars at the V1 and V3 stages for the presence of isoflavones and their glycosides. The analysis revealed that soybean leaves contain genistein and its glycoside genistin as the predominant isoflavonoids in the leaf tissue [20]. The soybean cultivars AC Colombe, Conrad, AC Glengarry, OAC Avatar, and Maple Arrow were considered low; OAC Lakeview, OAC Wallace, Beer Friend, and RCAT 1101a were considered moderately high; and OT06-22, Pagoda, Harosoy 63, and OT06-23 were considered high for leaf isoflavonoid levels averaged between the V1 and V3 stages. Based on their leaf isoflavonoid levels, soybean cultivars were chosen for aphid trials. These included nine Ontario-grown cultivars and two control types: aphid resistant E16 265, E16 266, and E16 267, and aphid susceptible Beer Friend.

After the 10-day trial beginning at V3, the leaf isoflavonoid levels were measured again. The results revealed that the cultivars which consistently had the highest concentration of total isoflavonoids were RCAT 1101a, OT06-22, Harosoy 63, and the aphid-resistant control lines E16 265, E16 266, and E16 267. In most cultivars, the control (no herbivore) leaves had the same isoflavonoid levels as the corresponding aphid-treated leaves in both the third and top leaves, with the exception of Maple Arrow and Beer Friend (Figure 1). In the case of Beer Friend, the aphid-infested top leaves had significantly less isoflavonoids than the control leaves (three-way ANOVA [cultivar × treatment × leaf]; d.f. = 12, 104; F = 5.063; *p* < 0.0001), while the Maple Arrow control top leaves were found to have no detectable isoflavonoids. Since the remainder of the cultivars were found to have no significant difference in isoflavonoid levels across the four sets of leaves that were analysed, the mean concentration of those leaves indicated that five cultivars had higher levels than the others (three-way ANOVA [cultivar]; d.f. = 12, 104; F = 31.761; *p* < 0.0001), with AC Colombe, Conrad, and Beer Friend having the lowest levels. Although the isoflavonoid levels did not change in response to aphid infestation in both the third and top leaves of the soybean cultivars included in the present study, the levels of isoflavonoids in the aphid resistant lines E16 265, E16 266, and E16 267 and the susceptible line ‘Beer Friend suggest that these compounds may be involved in conferring resistance.

From an initial number of two aphid nymphs placed on each third full leaf, a wide range of aphid survival and fecundity was observed across the 13 soybean cultivars tested. The fewest aphid nymphs and adults (<5/3rd leaf) were observed on the aphid-resistant lines, E16 265, E16 266, and E16 267, as well as the Pagoda, RCAT 1101a, and Beer Friend cultivars (Kruskal-Wallis; d.f. = 12; chi-sq. = 40.451; *p* < 0.0001) relative to the Conrad, OAC Wallace, and Harosoy 63 cultivars (Figure 2). The cultivars that were more resistant to aphids generally had the highest isoflavonoid concentrations in the third leaf compared to the other cultivars after 10 days. In contrast, regression analysis determined a significant, moderately negative correlation (Pearson log-rank; r = −0.309; d.f. = 50; t = −2.2962; *p* = 0.0259) between isoflavonoid concentration and aphid numbers when all 13 cultivars were included. 

### 3.2. Relative Aphid Resistance Based on Isoflavonoid Levels in Soybean Cultivars—4-Week Trial

Similar to the 10-day trial, soybean cultivars at the V1 stage were used for a 4-week aphid infestation trial. After 4 weeks, leaf samples were collected and isoflavonoid levels were measured. One of the initial 12 cultivars, OAC Strive, did not grow well over the 4-week period in the greenhouse, and thus was not included in the comparison. The results demonstrated that most of the 11 soybean cultivars with and without aphids (controls) were found to have the same concentration of total isoflavonoids in the top three trifoliate leaves, with the exception of AC Colombe and OAC Avatar, where the aphid-infested leaves had significantly higher isoflavonoid levels (two-way ANOVA [cultivar x treatment]; d.f. = 10,43; F = 2.521; *p* = 0.017) (Figure 3). When the mean isoflavonoid concentrations from the aphid-free and aphid-infested top trifoliates were compared, two cultivars, Maple Arrow and Harosoy 63, had significantly higher concentrations than those cultivars with the lowest concentrations (two-way ANOVA [cultivar]; d.f. = 10,43; F = 29.28; *p* < 0.0001), including AC Glengarry, OAC Lakeview, and Conrad.

The more resistant cultivars can be listed in order from the fewest to the most aphids counted as follows: OT06-22 < OT06-23 < Harosoy 63 < OAC Avatar (Figure 4). There were significantly fewer aphids on the top three trifoliate leaves of these cultivars (two-way ANOVA [cultivar]; d.f. = 10,66; F = 279.799; *p* < 0.0001) compared to the remaining cultivars, with the exception of AC Colombe (*p* > 0.05). There was a significant interaction between the cultivar and trifoliate (two-way ANOVA [cultivar x trifoliate]; d.f. = 20.66; F = 6.439; *p* < 0.0001), and two of the more susceptible cultivars (Maple Arrow and Pagoda) had significantly fewer aphid numbers on the first (top) trifoliate compared to the second and/or third leaves (*p* < 0.0001). A regression analysis of isoflavonoid concentrations in the first, second, and third trifoliates and the total aphid counts on each set of trifoliate leaves determined a non-significant, positive correlation (Pearson log-rank; r = 0.320; d.f. = 30; t = 1.9143; *p* = 0.06516).

The damage to soybean leaves by aphid feeding was assessed with the top three trifoliate leaves in eight cultivars (Figure 5), as images from three of the original eleven cultivars were lost. The percent damage to the leaves was determined using scanned images of the upper leaf surface of the aphid-infested plants relative to leaves from the control plants. The mean percent leaf damage caused by soybean aphids on all three trifoliate leaves was low (<6%) for most of the cultivars, with the exception of AC Glengarry (14.7%), which had significantly greater damage than all of the other seven cultivars (two-way ANOVA [cultivar]; d.f. = 7.47; F = 73.340; *p* < 0.0001). We also determined a significant interaction between trifoliate and cultivar for aphid tolerance (two-way ANOVA [cultivar x trifoliate]; d.f. = 14.47; F = 18.242; *p* < 0.0001). For the three less tolerant cultivars, the amount of leaf damage was greater (*p* < 0.05) on the first (top) trifoliate compared to the second and third trifoliate leaves for AC Glengarry and OAC Avatar, whereas in Harosoy 63, the damage was greater (*p* < 0.05) on the third leaf than the first and second. Regression analysis for isoflavonoid concentration and aphid leaf damage indicated that the relationship was not significant, and the correlation was low and negative (Pearson log-rank; r = −0.198; d.f. = 22; t = −0.9473; *p* = 0.3538). Generally, there was a low correlation between the resistance to aphids and total isoflavonoids, indicating that these compounds did not explain the range of resistance observed to soybean aphids. However, the 4-week trial findings do suggest different resistance ratings for the studied cultivars based on the aphid numbers per leaf (antibiosis) and the amount of leaf damage (tolerance) observed (Table 1). Based on these ratings, further leaf chemical analyses were applied to identify other potential metabolite differences.

### 3.3. Metabolomic Analyses

Since leaf isoflavonoid concentrations did not show a strong relationship with aphid resistance, other phytochemicals potentially involved in soybean aphid resistance were explored using metabolomic profiling of the resistant (R) and susceptible (S) cultivars, and other cultivars that were partially resistant or tolerant (T) (Table 1). A targeted and non-targeted approach using high-resolution LC-MS was applied to leaf extracts prepared from the 4-week greenhouse trials, and metabolite profiles were compared for the different cultivars collected with and without aphid feeding. In the absence of aphid infestation, there was no significant clustering between R, T, or S cultivars (Figure 6a). Conversely, with aphid infestation, a clear separation based on the metabolite profile was observed between the R cultivars and the T and S cultivars (Figure 6b). Differential analysis was used to determine the specific metabolites that were either increased or decreased in response to aphid infestation across the R, T, and S cultivars (Appendix A). In the R cultivars, only two and 11 features were significantly (FDR < 0.05) decreased or increased, respectively, upon aphid infestation. A much greater number of features were significantly altered by infestation in the T cultivars: two were decreased and 76 were increased. This indicates that 9.5% of the detected features in T cultivars were significantly altered. In the S cultivars, seven features were significantly decreased while 22 were increased. In all three classes of cultivars, a feature with the formula C_21_H_22_O_10_ was significantly increased upon infestation. Based on MS/MS, this feature was confirmed to possess a glucose moiety and has a formula that corresponds with dihydrogenistin, derived from dihydrogenistein. Methyladenosine, a nucleoside, was also found to be increased in T and S cultivars upon aphid infestation. Similar to what is shown in the PCA plots (Figure 6), there were no significantly different metabolite features between R and S cultivars in the absence of aphids (Appendix A). Conversely, a large perturbation of metabolite levels was observed between R and S cultivars in the presence of aphids (Appendix A).

Free amino acids (FAAs), the biogenic amines tyramine and histamine, and some isoflavonoids were quantified in the leaf material (Table 2). Important differences between the FAA levels and two aglycones (daidzein and genistein) were also found to vary in leaf concentration across the five cultivars (two-way ANOVA; d.f. = 4.19; F > 5.7648; *p* < 0.0033). One of the R aphid-infested soybean cultivars (Harosoy 63) had significantly higher levels of daidzein (two-way ANOVA; d.f. = 4.19; F = 4.1632; *p* = 0.013761) compared to the other R cultivar (OAC Avatar; Tukey’s test; *p* < 0.0001), as well as the T (Conrad; Tukey’s test; *p* < 0.0001) and the S (Maple Arrow; Tukey’s test; *p* = 0.0005) cultivars. This indicated that the isoflavonoids analyzed by high-resolution LC-MS were still found to have a low correlation with aphid resistance. Two glycosides (daidzin and glycitin) were also detected; however, their concentrations did not differ (two-way ANOVA; d.f. = 4.19; F = 1.5429; *p* > 0.2301) between the control and aphid-infested cultivars. In the R cultivars, FAA concentrations were not induced by aphid feeding any more than the FAA levels in the T and S cultivars (Table 2). Only in one case was the effect of aphid feeding associated with an increased FAA (Lys) concentration relative to the control leaves of the R cultivar OAC Avatar (two-way ANOVA; d.f. = 4.19; F = 21.595; *p* < 0.0001), while Phe (*p* = 0.0078), Try (*p* = 0.0886), and Gly (*p* < 0.0001) were significantly less (two-way ANOVA; d.f. = 4.19; F < 11.2803) in the aphid-infested leaves relative to the controls. In no cases were the FAA levels different between aphid-infested and control leaves for the most resistant cultivar, Harosoy 63. In both the T cultivars, six and eight FAAs, respectively, had significantly greater concentrations in the leaves of aphid-infested versus control leaves; however, of these, only five FAAs were the same (Gln, Gly, Histamine, Arg, and Lys; two-way ANOVA; d.f. = 4.19; F < 21.595; *p* < 0.0001). The S cultivar also had six FAAs that increased after aphid feeding, and three were the same as those found to change in the T cultivar, Pagoda (Gln, Asn, and Lys; two-way ANOVA; d.f. = 4.19; F > 3.5159; *p* < 0.0261). The differences in FAAs between the R cultivar, Harosoy 63, and the S cultivar on aphid-infested leaves were associated with significantly lower concentrations (two-way ANOVA; d.f. = 4.19; F > 4.2356; *p* < 0.0128) of Met, Tyr, Gln, and His in the R leaves. The same R cultivar also had significantly lower FAA levels than the T cultivars, Conrad (Met, Ala, Gln, Ser, and Lys; two-way ANOVA; d.f. = 4.19; F > 4.7915; *p* < 0.0077) and Pagoda (Ala, Gly, Gln, Asn, Glu, Asp, His, Arg, and Lys; two-way ANOVA; d.f. = 4.19; F > 3.5159; *p* < 0.0261). OAC Avatar (R) also had significantly lower levels (two-way ANOVA; d.f. = 4.19; F > 11.2803; *p* < 0.0001) of Gly and Lys than the T cultivar, Pagoda. The levels of the biogenic amines tyramine and histamine were significantly greater (two-way ANOVA; d.f. = 4.19; F > 14.085; *p* < 0.0001) in the S cultivar than the R cultivar, Harosoy 63, and histamine concentrations were induced (two-way ANOVA; d.f. = 4.19; F = 14.753; *p* < 0.0001) in both the T, Pagoda (Tukey’s test; *p* < 0.0001), and S cultivar (Tukey’s test; *p* < 0.0001) by aphid feeding.

## 4. Discussion

The present study focused on the antibiosis resistance and tolerance of MG II Ontario cultivars to biotype 1 soybean aphids. Upon aphid infestation to soybean cultivars, differences in leaf damage were observed, which may have been due to the characteristics of antibiosis, antixenosis, or both in each cultivar. Antibiosis could be the reason for the relatively slow growth of a population on a resistant cultivar compared to that of a more susceptible cultivar, which results in a reduced amount of damage [7]; however, antixenosis was more likely the cause when the numbers of aphids per leaf were not statistically different yet leaf damage varied between cultivars. In other studies [27,28], high leaf damage from aphid feeding, as exhibited by AC Glengarry and OAC Avatar cultivars, was associated with hypersensitive response (HR)-like symptoms characterized by necrosis and chlorosis (control and aphid-infested leaves from both cultivars are shown in Appendix A). Tolerance to aphids was observed with some of the cultivars tested, in particular Conrad and Pagoda. The numbers of aphids (Figure 4) were high enough to cause significantly greater or at least similar amounts of feeding damage as with other soybean cultivars (AC Glengarry and OAC Lakeview); however, percent damage was actually less (Figure 5).

As shown in Figure 1, R cultivars generally accumulated more isoflavonoids compared to S cultivars within a period of 10 days post-V3 stage. The data also show that these compounds are not induced upon aphid infestation. Despite the fact that isoflavonoid levels did not increase in response to aphid attack, it cannot be ruled out that the higher basal level of these compounds in resistant genotypes contributed to this resistance. However, the levels of isoflavonoids in the R leaves were no greater than those in the S leaves by 4 weeks post-V1 stage. The findings also indicated that the selected leaf isoflavonoids analyzed by liquid chromatography were not strongly correlated with antibiosis resistance for soybean aphids, or with tolerance to leaf damage.

The Ontario soybean cultivars surveyed for aphid resistance indicated differences in the total isoflavonoid levels in the top leaves sampled at the early V1 and V3 growth stages. The cultivars were ordered into three groups representing low, moderately high, or high total isoflavonoid leaf concentrations for the resistance screening. At the end of the 4-week greenhouse trial, three of four cultivars (OT06-22, OT06-23, and Harosoy 63) that had the highest V1 isoflavonoid concentrations were found to have the fewest aphids, while two of four cultivars with the lowest isoflavonoid concentrations (Maple Arrow and Conrad) had the highest aphid numbers. However, no strong relationship was determined when the aphid counts and leaf isoflavonoid data from all 11 cultivars were correlated. These findings suggest that isoflavonoid levels in the early growth stages are only predictive of aphid resistance or susceptibility when the concentrations in the leaves are either in the high or low levels, respectively. Another theory could be that isoflavonoids are more important defences when plants are first colonized by aphids, as aphids can adapt over several generations to the constitutive levels, leading to the cultivars being considered susceptible by later growth stages. Or perhaps other factors, such as nutrition, rather than isoflavonoid levels are more important at the later growth stages. In the cases where the same cultivar was used in both the 10-day and 4-week trial, differences in the relative resistance rating were observed. For example, the Pagoda cultivar was considered as aphid resistant based on the lower number of nymphs produced over 10 days on the third full leaves (Figure 2), while it was one of the more susceptible cultivars when compared at the end of the 4-week period (Figure 4). Similarly, Harosoy 63 was one of the more aphid-susceptible cultivars in the shorter duration day trial (Figure 2), but was considered less susceptible to aphids after 4 weeks on the whole plant (Figure 4). This was described as density-dependent aphid resistance [29]. It has been recommended that aphid counts on the whole plant not be used over short durations (e.g., over 10 days from inoculation) since most aphids do not move far from the inoculated trifoliate during that period [5]. It was suggested instead to count aphids on the whole plants in the late vegetative or early reproductive stages to determine the resistant accessions. Others have found that the same genotypes can differ in resistance based on the growth stage at the start and the extent of the plant included in the observations at the end of the trial. For example, cultivars in trials that began at the V3 stage showed greater resistance than those that began at V1 [7], and differences in the resistance to aphids of several biotypes were noted between whole plants and detached leaves from those plants [6].

Another finding from the present work was that aphid feeding had little effect on isoflavonoid levels, regardless of whether it was a cultivar considered to be resistant or susceptible to aphids. In only two cases did aphid feeding significantly alter isoflavonoid levels: AC Colombe and OAC Avatar, both in the lower isoflavonoid category, had significantly higher isoflavonoids on aphid-infested leaves compared to the controls after 4 weeks (Figure 3). A similar study where the length of aphid infestation was 3 weeks showed significant increases in the leaf levels of the isoflavones daidzein, genistein, and formononetin [17]. In contrast, transcriptome analysis found there were no differences between aphid-infested versus un-infested resistant plants with the *Rag1* gene after 21-days, and indicated the resistant soybean lines had higher constitutive defenses than the susceptible lines. In *Glycine soja* Siebold and Zucc., resistance has been correlated with the daidzein content in its leaves, which produces growth inhibitory effects with the Oriental leafworm *Spodoptera litura*. [30]. Similarly, *S. litura* herbivory of soybean (cv. Tamahomare) led to induced levels of daidzin, 4′,7-dihyroflavone, daidzein, and formononetin [13]. In the present study, daidzein and genestein were the only isoflavone aglycones detected by high-resolution LC-MS. Both isoflavones were significantly greater in only one of the two R cultivars (Harosoy 63) compared to the S cultivar (Table 2), suggesting that isoflavonoids are not solely responsible for the observed aphid resistance.

Mass spectrometry analyses detected other metabolite differences between the R, T, and S cultivars with aphid feeding, mainly associated with free amino acids (FAAs). The FAA differences are linked with nutrition, and may be the cause of lower aphid feeding and associated damage observed in the resistant cultivars. An early study on the relevance of FAA quality and quantity and soybean resistance associated reduced nutritional quality with the *Rag1* gene [31]. The FAAs with higher concentrations in the aphid-susceptible lines at the V6 and R2 stages were Aba, Asn, Gln, Glu, His, Pro, and Ser. Our results were similar, as Gln and His were also significantly lower in the R versus S cultivar, yet different because Met, Tyr, and Asn were significantly lower in Harosoy 63. In comparison, another hemipteran, *Megacopta cribraria* (F.) (Plataspidae), was found to have higher nymph production when the concentrations of Asp, Try, Ala, Phe, and Ser were greater in soybean leaves; however, production was less when Leu and Thr levels were higher [32]. This indicates the effects that FAA type and quantity can have on soybean herbivores.

We observed greater FAA induction from soybean aphid feeding in the T and S cultivars, where five amino acids (Gln, Gly, Histamine, Arg, and Lys) were found in common. No aphid induction of FAAs was observed in the R cultivar, Harosoy 63, and only Lys was induced in OAC Avatar. Another explanation for the difference in the number and amount of FAAs induced between R, T, and S cultivars may be due to the aphid pressure, as was noted by Florencio-Ortiz et al. [33]. In their work, a higher density of green peach aphids (*M. persicae*) on pepper (*Capsicum annuum*) leaves produced a greater increase in total FAA content, including Phe, Tyr, Trp, Val, Ile, Leu, Arg, Lys, Met, Thr, Ala, Asn, and His, compared to a low aphid density. The main mechanism responsible for FAA induction by phloem-feeding insects, such as aphids, is the redirection of nutrients from other plant tissues, which in turn produces increased enzyme activity in amino acid metabolism, including glutamine synthetase, lysine decarboxylase, and proteases [34].

Metabolic studies have also uncovered compounds other than amino acids and isoflavonoids that are associated with resistance to other soybean herbivores [20,35]. Soybean genotypes containing higher levels of the flavonoid methylquercetin and other glycoconjugates were more resistant to caterpillars (*Anticarsia gemmatalis* Hubner) [35], while an unidentified peptide-like compound was found in higher amounts in spider mite (*Tetranychus urticae* Koch)-infested soybean resistant to the mites [20]. The biogenic amine tyramine, which was highest in the S cultivar aphid-fed leaves, has been associated with reduced probing by bird cherry-oat aphids (*Rhopalosiphum padi* L.) [36]. In the present study, tyramine leaf concentrations may not have reduced feeding behaviour sufficiently to decrease the leaf feeding damage by large numbers of soybean aphids.

The selection of currently available soybean aphid resistant cultivars can assist growers to minimize yield losses and reduce the amount of insecticide applied during the season. Future research should evaluate the promising cultivars identified as resistant to aphids, including OT06-22, under field conditions and determine the metabolite profiles. The difference in metabolites between susceptible and resistant cultivars can be applied both as biomarkers to predict cultivars resistant to other pests and herbivores and for use in breeding programs.

## Figures and Tables

**Figure 1 insects-13-00356-f001:**
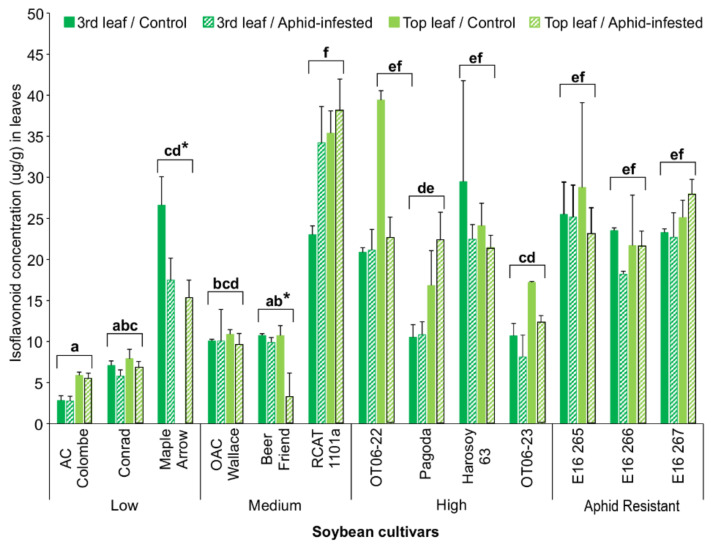
Mean isoflavonoid concentrations (µg/g) (±S.E.) for 3rd and top full leaves from 13 soybean cultivars at the end of the 10-day bioassay. Low, mid, and high indicate the leaf isoflavonoid levels of the soybean cultivars at the V1 and V3 stages as determined in Scott et al. [20]. Aphid-resistant lines were used as controls. Bars with the same lowercase letters indicate no significant difference (*p* > 0.05) for the average isoflavonoid concentrations between those cultivars. Bars with an asterisk (*) indicate a significant difference (*p* < 0.05) between the control and aphid-infested leaves for that cultivar.

**Figure 2 insects-13-00356-f002:**
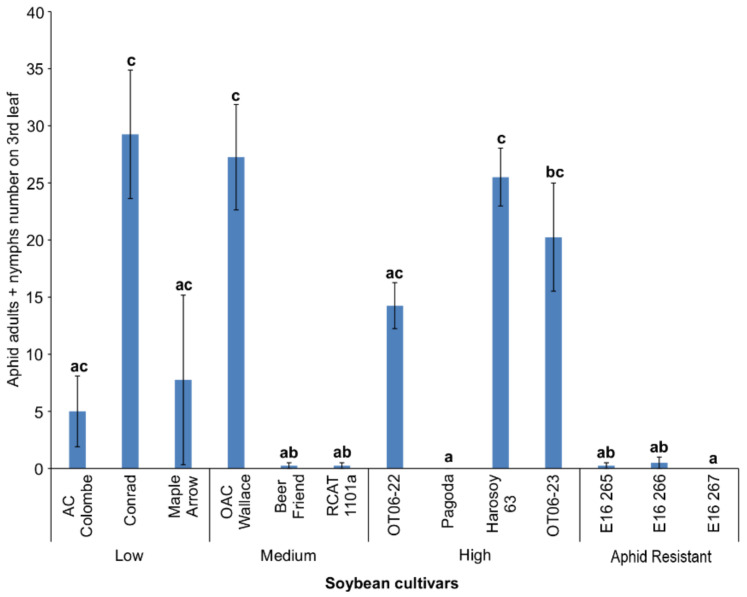
Mean aphid adult and nymph (±S.E.) counts on the 3rd full leaf of 13 soybean cultivars after 10 days of growth. Bars with the same lowercase letters indicate no significant difference (*p* > 0.05) for the average aphid count between those cultivars.

**Figure 3 insects-13-00356-f003:**
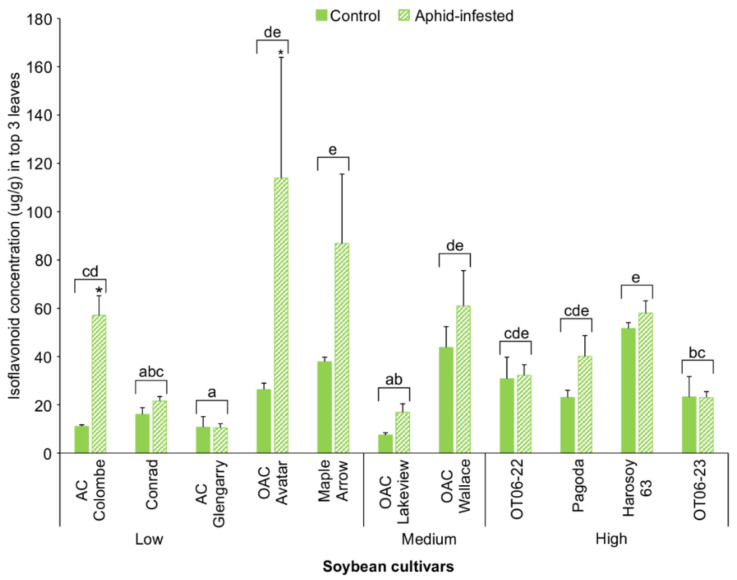
Mean isoflavonoid concentrations (µg/g; ±S.E.) in combined top three trifoliate leaves with and without aphids of 11 soybean cultivars after 4 weeks growth under greenhouse conditions. Bars with the same lowercase letters indicate no significant difference (*p* > 0.05) for the isoflavonoid concentrations between those cultivars. Bars with an asterisk (*) indicate a significant difference (*p* < 0.05) between the control and aphid-infested leaves for that cultivar.

**Figure 4 insects-13-00356-f004:**
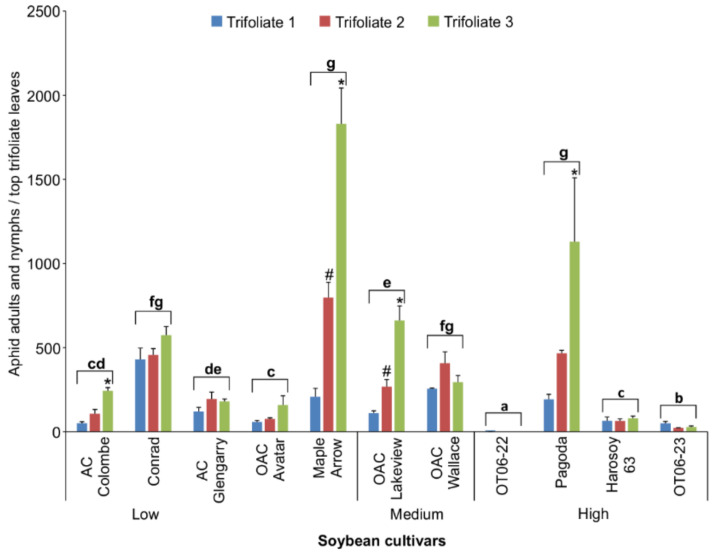
Mean aphid counts (±S.E.) on top 3 trifoliate leaves for 11 soybean cultivars after 4 weeks growth in the greenhouse. Bars with the same lowercase letters indicate no significant difference (*p* > 0.05) for the average aphid counts between those cultivars. Bars with an asterisk (*) or hashtag (#) indicate a significant difference (*p* < 0.05) between the number of aphids on the 1st and 3rd or 1st and 2nd trifoliate leaves, respectively.

**Figure 5 insects-13-00356-f005:**
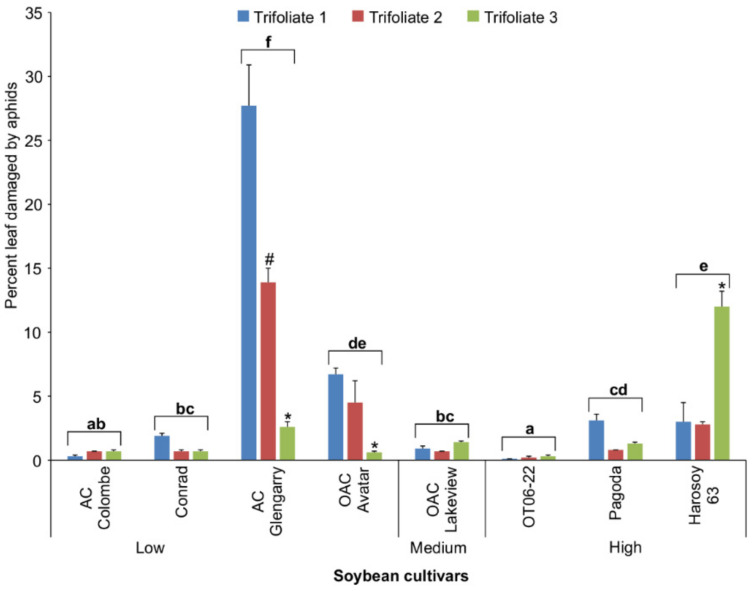
Mean percent damage (±S.E.) by soybean aphids to the top 3 trifoliate leaves for 8 soybean cultivars after 4 weeks growth in the greenhouse. Bars with the same lowercase letters indicate no significant difference (*p* > 0.05) for mean percent leaf damage between those cultivars. Bars with a hashtag (#) or asterisk (*) indicate a significant difference (*p* < 0.05) between the percent damage on the 2nd versus the 1st and 3rd, or the 3rd leaf versus the 1st and 2nd trifoliate leaves, respectively.

**Figure 6 insects-13-00356-f006:**
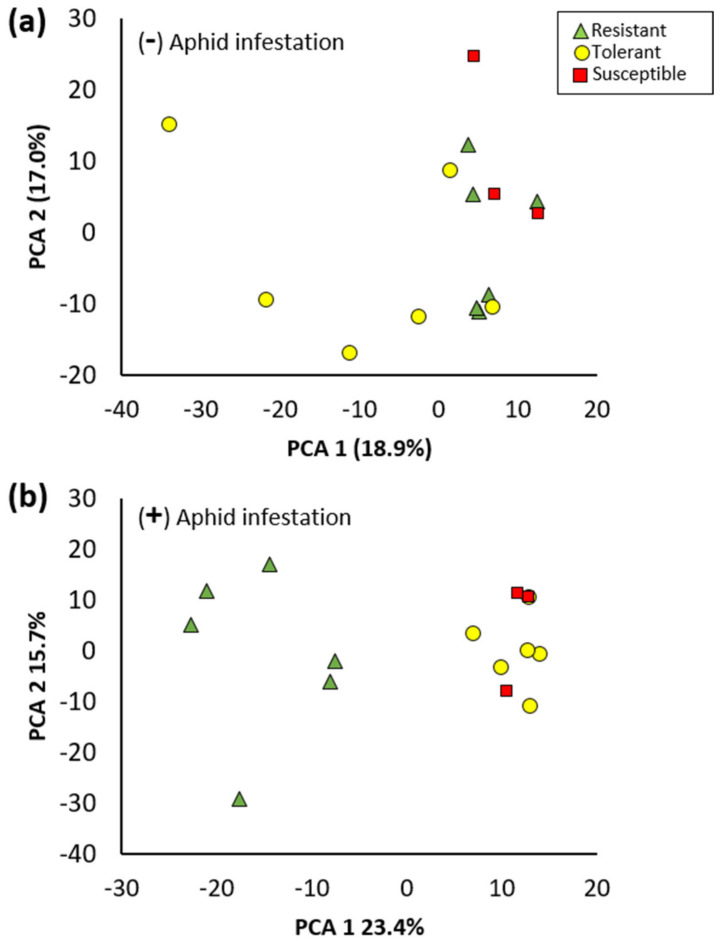
PCA plots derived from metabolite features in the (**a**) absence of aphids and (**b**) presence of aphids.

**Table 1 insects-13-00356-t001:** Relative aphid resistance ratings for five Ontario-grown soybean cultivars.

Soybean cv.	Characteristic	Aphid Resistance	Rating ^1^
Maple Arrow	Higher numbers of aphids at week 4	Susceptible	S
Conrad	Higher numbers of aphids at week 4, but relatively low leaf damage	Susceptible and partially tolerant	T
Pagoda	Higher numbers of aphids at week 4, but relatively low leaf damage	Susceptible and partially tolerant	T
Harosoy 63	Lower numbers of aphids at week 4, but moderate leaf damage	Less susceptible but not as tolerant	R
OAC Avatar	Lower numbers of aphids, but moderate leaf damage	Resistant but not as tolerant	R

^1^ S—susceptible at late stage; T—tolerant at late stage; R—resistant at late stage.

**Table 2 insects-13-00356-t002:** The effect of soybean aphid feeding on the concentrations of isoflavones, amino acids, and other bioactive compounds in the leaves of one susceptible, two resistant, and three tolerant soybean cultivars after 4 weeks.

	Resistant Varieties								Tolerant Varieties								Susceptible Variety			
Metabolites	Harosoy 63				OAC Avatar				Conrad				Pagoda				Maple Arrow			
	Control		Aphid		Control		Aphid		Control		Aphid		Control		Aphid		Control		Aphid	
	^1^ Avg	(SE)	Avg	(SE)	Avg	(SE)	Avg	(SE)	Avg	(SE)	Avg	(SE)	Avg	(SE)	Avg	(SE)	Avg	(SE)	Avg	(SE)
Genistein	6112	543 ^ab^	12,551	6385 ^a^	1432	862 ^b^	823	410 ^b^	346	163 ^b^	86.6	50.9 ^b^	3350	934 ^ab^	2121	627 ^ab^	2301	650 ^ab^	1207	366 ^b^
Daidzein	0.8	0.1 ^ab^	1.4	0.1 ^a^	0.0	0.0 ^c^	0.1	0.0 ^c^	0.3	0.3 ^bc^	0.0	0.0 ^c^	0.5	0.1 ^bc^	0.9	0.0 ^ab^	0.0	0.0 ^c^	0.4	0.1 ^bc^
Daidzin	7.5	3.3 ^b^	12.8	8.4 ^ab^	5.3	2.8 ^b^	25.3	10.4 ^ab^	3.7	1.6 ^b^	8.3	0.7 ^ab^	10.7	4.1 ^ab^	27.0	5.0 ^ab^	9.2	6.3 ^ab^	48.4	18.5 ^a^
Glycetin	1.0	0.5 ^a^	2.7	1.7 ^a^	1.4	0.9 ^a^	4.1	1.4 ^a^	1.9	0.8 ^a^	2.8	0.4 ^a^	1.3	0.6 ^a^	4.2	0.0 ^a^	1.4	0.8 ^a^	6.9	2.9 ^a^
^2^ Tyram.	3.1	0.6 ^bc^	3.8	0.9 ^bc^	2.9	0.7 b^c^	1.8	0.4 ^c^	0.5	0.1 ^c^	1.2	0.1 ^c^	2.0	0.6 ^c^	1.3	0.0 ^c^	5.7	1.3 ^ab^	8.5	1.2 ^a^
Phe	43.1	9.0 ^ab^	30.0	5.4 ^ab^	107.9	42.9 ^a^	11.6	3.0 ^b^	15.3	7.0 ^b^	34.8	5.6 ^ab^	35.5	3.1 ^ab^	65.5	2.1 ^ab^	36.2	15.0 ^ab^	31.0	9.3 ^ab^
Try	92.1	8.3 ^ab^	83.2	21.2 ^ab^	102.0	26.0 ^a^	16.9	3.0 ^b^	15.7	3.8 ^b^	18.8	3.5 ^b^	20.8	0.6 ^ab^	12.4	0.1 ^b^	78.7	10.7 ^ab^	64.2	33.9 ^ab^
Ile	24.6	6.6 ^a^	18.8	4.3 ^a^	44.1	15.8 ^a^	7.9	1.6 ^a^	18.1	7.9 ^a^	16.5	3.3 ^a^	13.3	2.9 ^a^	10.0	0.5 ^a^	25.8	13.5 ^a^	21.6	5.6 ^a^
Leu	40.6	1.7 ^ab^	31.0	6.5 ^ab^	65.4	17.6 ^a^	13.6	4.6 ^ab^	6.9	3.4 ^b^	33.4	9.5 ^ab^	19.8	2.6 ^ab^	21.4	0.7 ^ab^	38.4	21.1 ^ab^	46.9	12.2 ^ab^
Met	1.5	0.3 ^abc^	0.6	0.3 ^c^	0.4	0.3 ^c^	0.5	0.3 ^c^	0.8	0.5 ^c^	4.9	1.1 ^a^	1.0	0.3 ^bc^	3.3	1.1 ^abc^	1.9	1.1 ^abc^	4.1	0.6 ^ab^
Tyr	15.1	4.0 ^b^	8.6	1.9 ^b^	21.9	10.4 ^ab^	7.0	1.5 ^b^	7.8	1.1 ^b^	27.9	5.7 ^ab^	6.5	4.0 ^b^	48.3	8.9 ^ab^	26.9	11.8 ^ab^	62.5	17.9 ^a^
Val	627	123 ^a^	405	177 ^a^	664	65.5 ^a^	461	168 ^a^	323	72.8 ^a^	310	89.8 ^a^	400	94.3 ^a^	329	46.9 ^a^	711	157 ^a^	333	64.9 ^a^
Pro	245	37.0 ^a^	107	21.2 ^ab^	134	36.1 ^ab^	34.7	3.8 ^b^	26.5	3.7 ^b^	74.6	18.0 ^b^	53.8	11.2 ^b^	93.9	12.3 ^ab^	123	68.8 ^ab^	109	37.1 ^ab^
Ala	13.4	2.0 ^bc^	12.0	1.9 ^bc^	14.1	2.9 ^bc^	12.2	2.7 ^bc^	6.9	1.5 ^c^	43.0	9.1 ^a^	52.8	10.9 ^a^	48.8	0.5 ^a^	8.7	3.4 ^c^	38.7	8.0 ^ab^
Thr	77.8	19.1 ^a^	58.7	10.0 ^a^	89.1	9.6 ^a^	48.8	13.6 ^a^	39.4	7.8 ^a^	76.8	20.0 ^a^	129.4	34.2 ^a^	106.9	13.8 ^a^	80.5	14.7 ^a^	102.3	21.3 ^a^
Gly	2.3	0.7 ^bcd^	4.8	0.6 ^bcd^	7.9	1.3 ^abc^	0.9	0.3 ^d^	2.1	0.9 ^bcd^	8.8	0.1 ^ab^	3.7	0.8 ^bcd^	13.2	5.3 ^a^	1.2	0.5 ^cd^	8.1	2.0 ^ab^
Gln	34.7	7.7 ^c^	28.2	5.2 ^c^	71.7	2.3 ^bc^	34.8	7.4 ^c^	22.1	2.1 ^c^	113	13.1 ^ab^	50.9	6.1 ^c^	160	9.7 ^a^	46.3	12.1 ^c^	167	28.4 ^a^
Ser	103	10.0 ^bc^	57.5	16.1 ^cd^	114	10.3 ^abc^	56.4	8.3 ^cd^	26.5	5.5 ^d^	178	11.4 ^a^	116	10.3 ^abc^	110	18.0 ^abc^	125	18.0 ^ab^	119	16.6 ^abc^
Asn	18.6	4.4 ^c^	18.0	4.6 ^c^	23.5	1.4 ^c^	24.9	6.9 ^c^	11.4	0.8 ^c^	54.8	10.2 ^bc^	75.2	39.1 ^abc^	181	36.5 ^a^	25.5	8.0 ^c^	143	42.8 ^ab^
Glu	841	188 ^bc^	681	30.3 ^c^	1440	92.4 ^abc^	654	73.5 ^c^	660	318 ^c^	646	36.8 ^c^	1475	169 ^abc^	1999	133 ^a^	1605	312 ^ab^	766	132 ^bc^
Asp	478	44.9 ^bc^	406	74.5 ^c^	649	63.0 ^bc^	596	156 ^bc^	282	132 ^c^	239	32.6 ^c^	804	204 ^bc^	2403	127 ^a^	1024	189 ^b^	512	64.3 ^bc^
^3^ Histam.	0	0 ^b^	0	0 ^b^	0	0 ^b^	0.2	0.1 ^b^	0	0 ^b^	0.5	0.1 ^b^	0	0 ^b^	2.1	0.3 ^a^	0	0 ^b^	2.4	0.6 ^a^
His	9.9	1.4 ^c^	14.3	2.2 ^c^	28.1	1.4 ^c^	21.0	7.3 ^c^	6.5	1.4 ^c^	69.6	7.5 ^bc^	17.6	1.0 ^c^	237	7.7 ^a^	42.9	19.8 ^bc^	99.8	31.9 ^b^
Arg	19.9	3.1 ^b^	27.0	3.5 ^b^	27.6	7.5 ^b^	34.4	10.0 ^b^	26.1	4.7 ^b^	93.9	19.3 ^b^	18.4	2.8 ^b^	264	42.5 ^a^	20.4	4.0 ^b^	214	38.6 ^a^
Lys	10.5	2.2 ^c^	12.2	1.3 ^c^	16.0	2.3 ^c^	29.4	8.1 ^b^	28.3	10.2 ^c^	95.8	4.3 ^b^	21.7	3.8 ^c^	179	4.0 ^a^	16.1	3.9 ^c^	135	25.7 ^ab^

^1^ Mean concentrations (ng/g) ± standard error (SE) were not corrected for extraction recovery or SSE%; ^2^ Tyramine; ^3^ Histamine. Metabolites per row with the same letters are not statistically different (two-way ANOVA, *p* > 0.05).

## Data Availability

This data has been submitted to the NIH Common Fund’s National Metabolomics Data Repository (NMDR) website, the Metabolomics Workbench, https://www.metabolomicsworkbench.org (accessed on 17 February 2022).

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
