# Peer review of "Investigation of Metabolic Resistance to Soybean Aphid (Aphis glycines Matsumura) Feeding in Soybean Cultivars"

_insects, 2022, doi:10.3390/insects13040356_

Round 1

Reviewer 1 Report

The authors aimed to test and correlate levels of metabolites such as isoflavonoids, free amino acids, and other amines with soybean antibiosis resistance and tolerance to soybean aphid. The subject matter is interesting and the results could help advise breeding efforts for improved aphid resistance. Unfortunately, there are some problems with the methodology and sample size may not be large enough for some of the control groups.

Points of major concern:

  • Figure 2: Beer Friend is your susceptible control yet the aphid populations on it are reflective of a resistant phenotype. Did you expect this, and if not, why do you suppose this occurred? Was there a problem with the infestation? How many times was the entire 10d experiment replicated (did you see low infestation rates for Beer Friend among all replicated experiments)?

  • Lines 149-151 (10d experiment): I’m concerned about replication in your control plants for isoflavonoid quantification. You mention that for each cultivar you used one control plant (no treatment at all) and one plant with the “setup” (i.e. lanolin, paper clips, stake, pipe cleaners) but no aphids. Was only one of these plants used for isoflavone quantification (if so, how were statistics conducted)? Were both plants used (if so, you need to present evidence that the lanolin/paper clips/stake/pipe cleaners have no effect on isoflavonoid levels by conducting an additional set of experiments with no aphids but compare control vs “setup”)?

  • Line 159: the 4 week trial was done in three separate runs (4 cvs per run)
    • A) What measures were taken to ensure the varieties across these three trials could directly be statistically compared? This may be a confounding variable in your experiments if greenhouse conditions changed between trials.
    • B) Were 12 cultivars tested? I see results for 11?

Points to clarify:

  • Since metabolites were studied rather than genetic components, I suggest changing the word “genetic” in the manuscript title to “metabolic” to read, “Investigating soybean (Glycine max) metabolic resistance associated with anti-soybean aphid (Aphis glycines Matsumura) activity”

  • Lines 95-96: You state that aphids and stinkbugs encounter isoflavonoid compounds within the phloem. However, Hohenstein et al., 2019 (your reference 26) showed that isoflavones likely accumulate in soybean mesophyll cells (which would be during stylet probing stage) under soybean aphid infestation.

  • Line 148: “assess survival, growth and adult fecundity”. You reported only aphid populations, not data on survival or fecundity. I suggest replacing with “assess aphid population growth”

  • Lines 171-173/181-182/323-326: It’s unclear what the sample consisted of for the 4 wk isoflavone quantification. When I look at figure 3, it appears each sample consisted of one tube containing 3 leaves of one plant. Or did you collect and process each of the trifolates separately and then take an average of the three samples to generate figure 3?
    • (Along with the previous comment) Lines 241-244: 4 week experiment. It was unclear which aphid count you used to correlate with isoflavonoid levels; did you use total aphid numbers (e., sum of aphids on all trifoliates)?

  • Lines 267-268: “while the Maple Arrow control top leaves were found to have no measureable [sic] isoflavonoids possibly due to sampling or analysis errors.” Were these values reported as 0 and included in statistical analysis? Or treated as missing data?

  • There are several inconsistencies between the text and figures/tables (I’ll give just a few examples, but the authors should check other claims as well):
    • Lines 295-299: Text suggests that two cultivars (Maple Arrow and Harosoy 63) had significantly higher isoflavone levels than the cultivars with the lowest levels. However, the statistics indicated on Figure 3 show that OAC Avatar and OAC Wallace also have higher isoflavonoid concentrations than AC Glengarry/OAC Lakeview/Conrad.
    • Lines 407-408: Text suggests that only Phe decreased in OAC Avatar in response to aphid feeding yet Table 2 shows that Try and Gly also decrease in OAC Avatar.
    • Lines 413-415/Line 517: Text suggests that three (Gln, Asn, Lys) were altered by aphid infestation in both Maple Arrow (S cultivar) and Pagoda (T cultivar). However, Table 2 suggests Gln, Gly, Histamine, Arg, Lys are common between the two cultivars.
    • Line 534: You suggest that tyramine is induced by aphid feeding in the S cultivar but your data does not support this (not significant)

  • Could you clarify what you mean when you suggest (lines 438-439) that antixenosis is a likely cause for similar aphid population sizes causing vastly different leaf damage symptoms? Did you consider that the cultivars that have low aphid populations but high leaf damage (e.g., AC Glengarry, OAC Avatar) exhibit hypersensitive response (HR)-like symptoms characterized by necrosis and chlorosis? (e.g., see Villada et al., 2009 https://doi.org/10.1093/jxb/erp163 or Klingler et al., 2009 https://doi.org/10.1093/jxb/erp244). I suggest including sample images of the plant damage sustained, to help distinguish between HR and general aphid feeding damage.

  • Lines 461-463: “These findings suggest that isoflavonoid levels in early growth stages are only predictive of aphid resistance or susceptibility at either end of the range of measured leaf concentrations.” Can you please clarify what you mean by this?

Other thoughts:

  • Authors should be cognizant that if the colony (source of aphids) soybean variety has mid-to-high isoflavonoid levels that the aphids may have upregulated detoxification mechanisms, thereby may be better prepared to handle cultivars with higher isoflavonoid levels. Thus, your resulting effect size may be diminished by using Beer Friend as the colony variety.
  • Lines 469-471: Could be related to density-dependent resistance. See Natukunda et al., 2019 (https://doi.org/10.1093/jee/toz017).

Grammar/spelling/flow should be checked:

  • Resistant cultivar name spacing inconsistent in paper (i.e., “E16265” vs “E16 265”)
  • In some places, writing style is difficult to follow
  • Several minor spelling or formatting errors throughout manuscript (including Table 2)

Author Response

The authors aimed to test and correlate levels of metabolites such as isoflavonoids, free amino acids, and other amines with soybean antibiosis resistance and tolerance to soybean aphid. The subject matter is interesting and the results could help advise breeding efforts for improved aphid resistance. Unfortunately, there are some problems with the methodology and sample size may not be large enough for some of the control groups.

 Thank you for reviewing the manuscript and providing many suggestions for improvement. Please see the responses in red text and corrections in the manuscript revisions.

Points of major concern:

  • Figure 2: Beer Friend is your susceptible control yet the aphid populations on it are reflective of a resistant phenotype. Did you expect this, and if not, why do you suppose this occurred? Was there a problem with the infestation? How many times was the entire 10d experiment replicated (did you see low infestation rates for Beer Friend among all replicated experiments)?

Soybean cv. Beer Friend was not considered a susceptible control, rather it was included as part of the study since the soybean aphids had been maintained on BF for many generations prior to the beginning of the study. The 10 day trial on BF plants (and other lines) was replicated 4 times (4 separate plants/cultivar, four 10 day trials). There was no expectation that the aphids would be more or less susceptible to BF, rather that they would be adapted to it. This would not necessarily mean they would be more tolerant or resistant to BF compared to the other soybean lines cultivars that could be more susceptible or resistant. 

  • Lines 149-151 (10d experiment): I’m concerned about replication in your control plants for isoflavonoid quantification. You mention that for each cultivar you used one control plant (no treatment at all) and one plant with the “setup” (i.e. lanolin, paper clips, stake, pipe cleaners) but no aphids. Was only one of these plants used for isoflavone quantification (if so, how were statistics conducted)? Were both plants used (if so, you need to present evidence that the lanolin/paper clips/stake/pipe cleaners have no effect on isoflavonoid levels by conducting an additional set of experiments with no aphids but compare control vs “setup”)?

The control plants with and without the set-up were single replicates and were pooled together to represent the range of isoflavonoid leaf concentrations under both conditions. The average concentration and standard error for the both controls was then compared to the 4 replicate aphid-infested plants for each cultivar. The comparison of the control to aphid-infested plants allowed for an assessment of whether the aphids induced isoflavonoid levels above the control levels (both types). This point was added in the text (Lines 154-155).  

  • Line 159: the 4 week trial was done in three separate runs (4 cvs per run)
    • A) What measures were taken to ensure the varieties across these three trials could directly be statistically compared? This may be a confounding variable in your experiments if greenhouse conditions changed between trials.
    • B) Were 12 cultivars tested? I see results for 11?

The 4 varieties chosen for each of the 3 trials were selected based on the isoflavonoid leaf levels measured at the V1 stage in the original 18 cultivars that were screened. At least one representative cv. from each low, moderately high and high isoflavonoid grouping was included within each trial. The environmental conditions within the greenhouse over the course of the 3 months that the trials were conducted were held within the same temperature, humidity and lighting conditions. Of the 12 cultivars initially included in the 3 trials, one variety (OAC Strive – considered to have moderately high isoflavonoid levels) did not grow well up to the  4th week and was thus dropped from study.  This point was added to the text (Lines 300-301).   

Points to clarify:

  • Since metabolites were studied rather than genetic components, I suggest changing the word “genetic” in the manuscript title to “metabolic” to read, “Investigating soybean (Glycine max) metabolic resistance associated with anti-soybean aphid (Aphis glycinesMatsumura) activity”

Thank you for the suggestion – the title has been revised to read “Investigation of soybean (Glycine max) metabolic resistance to soybean aphid (Aphis glycines Matsumura) feeding” 

  • Lines 95-96: You state that aphids and stinkbugs encounter isoflavonoid compounds within the phloem. However, Hohenstein et al., 2019 (your reference 26) showed that isoflavones likely accumulate in soybean mesophyll cells (which would be during stylet probing stage) under soybean aphid infestation.

Thank you for pointing that difference out – the first paper cited examined the levels of IFS enzyme and gene expression in different soybean tissues. The fact that the leaf levels contain high isoflavonoid levels but low enzyme and gene expression was thought to be due to the transport of glycosylate isoflavonoids (daidzin) (more soluble) from the tissue of origin to the leaves, a fact which would contribute to their potential uptake by feeding aphids feeding in the phloem. However, the more recent findings by Hohenstein suggests that aglycones (daidzein) are found in the parenchyma and epidermal tissues close to the vasculature. This was added in the text (Lines 99-100).

  • Line 148: “assess survival, growth and adult fecundity”. You reported only aphid populations, not data on survival or fecundity. I suggest replacing with “assess aphid population growth”

 Thank you. This was revised in the text (Line 151).

  • Lines 171-173/181-182/323-326: It’s unclear what the sample consisted of for the 4 wk isoflavone quantification. When I look at figure 3, it appears each sample consisted of one tube containing 3 leaves of one plant. Or did you collect and process each of the trifolates separately and then take an average of the three samples to generate figure 3?

The method where 3 trifoliate leaflets was combined in one tube was correct (lines 175-179) – the single mid leaf from each trifoliate in each tube was not correct (Lines 189-191), these were removed from the text. Top, 2nd and 3rd trifoliate leaflets were analyzed, and the average was shown in Figure 3.

  • (Along with the previous comment) Lines 241-244: 4 week experiment. It was unclear which aphid count you used to correlate with isoflavonoid levels; did you use total aphid numbers (, sum of aphids on all trifoliates)?

The correlation was between the combined number of aphids on the 3 leaflets for the 3rd, 2nd and top  trifoliates and the respective isoflavonoid concentration measured in the combined leaflets. This was revised in the text (Lines 339-340).

  • Lines 267-268: “while the Maple Arrow control top leaves were found to have no measureable [sic]isoflavonoids possibly due to sampling or analysis errors.” Were these values reported as 0 and included in statistical analysis? Or treated as missing data?

That is a good point – the isoflavonoid levels for control Maple Arrow top leaf was given a value of 0, so despite the statement in line 267 that data was missing we decided to consider the concentration undetectable. A change was made in the text (Lines 276-277).  

  • There are several inconsistencies between the text and figures/tables (I’ll give just a few examples, but the authors should check other claims as well):
    • Lines 295-299: Text suggests that two cultivars (Maple Arrow and Harosoy 63) had significantly higher isoflavone levels than the cultivars with the lowest levels. However, the statistics indicated on Figure 3 show that OAC Avatar and OAC Wallace also have higher isoflavonoid concentrations than AC Glengarry/OAC Lakeview/Conrad.

Thank you. This was corrected in the text (Lines 309-310) to read “Maple Arrow and Harosoy 63, had significantly higher concentrations than the cultivars with the lowest concentrations (Two-way ANOVA [cultivar]; d.f.=10,43; F=29.28; P<0.0001),  AC Glengarry, OAC Lakeview and Conrad.”

  • Lines 407-408: Text suggests that only Phe decreased in OAC Avatar in response to aphid feeding yet Table 2 shows that Try and Gly also decrease in OAC Avatar.

The significantly lower Try and Gly were added in the text (Lines 423-425).

  • Lines 413-415/Line 517: Text suggests that three (Gln, Asn, Lys) were altered by aphid infestation in both Maple Arrow (S cultivar) and Pagoda (T cultivar). However, Table 2 suggests Gln, Gly, Histamine, Arg, Lys are common between the two cultivars.

Thank you. Yes the FAAs that should have been included were Gln, Arg, Gly, Lys, and Histamine. This was corrected in the text (Lines 429-430, 542).

  • Line 534: You suggest that tyramine is induced by aphid feeding in the S cultivar but your data does not support this (not significant)

This was corrected in the text to read “The biogenic amine, tyramine,  highest in the S cultivar aphid-fed leaves”,… (Line 559). 

  • Could you clarify what you mean when you suggest (lines 438-439) that antixenosis is a likely cause for similar aphid population sizes causing vastly different leaf damage symptoms? Did you consider that the cultivars that have low aphid populations but high leaf damage (e.g., AC Glengarry, OAC Avatar) exhibit hypersensitive response (HR)-like symptoms characterized by necrosis and chlorosis? (e.g., see Villada et al., 2009 https://doi.org/10.1093/jxb/erp163 or Klingler et al., 2009 https://doi.org/10.1093/jxb/erp244). I suggest including sample images of the plant damage sustained, to help distinguish between HR and general aphid feeding damage.

 Thank you. Yes we had not considered that possibility. A sentence was added in the text to read “In other studies (Villada et al. 2009; Klinger et al. 2009) high leaf damage from aphid feeding, as was exhibited by AC Glengarry and OAC Avatar, was associated with a hypersensitive response (HR)-like symptoms characterized by necrosis and chlorosis”. (Lines 458-460).

Images of control and aphid infested leaves were included in the supplementary Fig. S1 and S2.

  • Lines 461-463: “These findings suggest that isoflavonoid levels in early growth stages are only predictive of aphid resistance or susceptibility at either end of the range of measured leaf concentrations.” Can you please clarify what you mean by this?

This sentence was revised to read ““These findings suggest that isoflavonoid levels in early growth stages are only predictive of aphid resistance or susceptibility when the their concentrations in leaves are either in the low or high levels, respectively”. (Lines 485-486).

Other thoughts:

  • Authors should be cognizant that if the colony (source of aphids) soybean variety has mid-to-high isoflavonoid levels that the aphids may have upregulated detoxification mechanisms, thereby may be better prepared to handle cultivars with higher isoflavonoid levels. Thus, your resulting effect size may be diminished by using Beer Friend as the colony variety.

The authors agree, although the Beer Friend cv soybean plants were determined to have relatively low total isoflavonoid levels at growth stages later than V3 in comparison to Harosoy 63, Pagoda, Maple Arrow and OAC Avatar (data not shown in the manuscript). This is important to consider since the aphid colony was kept on BF plants that were generally older than V3 stage, thus they would have been exposed to relatively lower isoflavonoid leaf concentrations compared to the other cultivars.

  • Lines 469-471: Could be related to density-dependent resistance. See Natukunda et al., 2019 (https://doi.org/10.1093/jee/toz017).

 Thank you. A new sentence was added to the text to read: “This was described as “density-dependent aphid resistance”. (Lines 497-498)

Grammar/spelling/flow should be checked:

  • Resistant cultivar name spacing inconsistent in paper (i.e., “E16265” vs “E16 265”)

The spacing was corrected – now appears as E16 265, E16 266 or E16 267 in text.

  • In some places, writing style is difficult to follow

The authors have made an effort to improve the style by re-phrasing sentences – for example “Differences in leaf damage were observed due to characteristics of antibiosis, antixenosis or both for each cultivar” – changed to “Upon aphid infestation to soybean cultivars, differences in leaf damage were observed which may be due to characteristics of antibiosis, antixenosis or both for each cultivar”. (Lines 452-454)

  • Several minor spelling or formatting errors throughout manuscript (including Table 2)

Formatting errors in Table 2 were revised.

Reviewer 2 Report

The manuscript by Scott et al. investigated the relationship between isoflavonoids and free amino acids in soybeans with their resistance to aphids. They tested the concentration of isoflavonoids and free amino acids in 13 aphid resistant, tolerant and suscpetible soybean cultivars. They found that there was a low correlation between isoflavonoid and aphid resistance, but low level of free amino acids might increase the aphid resistance in soybeans. The data in this MS were well described, but there are still issues that need to be addressed. Below please find my detailed comments.

  • What’s the meaning of “low”, “medium”, “high” and “resistant” in Figure 1. Based on the description, “low”, “medium”, “high” and “resistant” seem to mean the concentration of isoflavonoids. But I am very confusing about the standard of “low”, “medium”, “high” and “resistant”. The concentration of isoflavonoids in RCAT1101a is the highest among the 13 soybean cultivars, but RCAT1101a belongs to “medium”. How to understand here?
  • Some data in Figure 4 and 5 are not consistent to each other. For example, the aphid counts on top 3 trifoliate leaves of AC Glengarry was low (Figure 4), but the percent damage by soybean aphids to the top 3 trifoliate leaves of AC Glengarry was the highest. How to understand here?
  • Line 14-15 and Line 33-36: Based on the data, I think this conclusion is not solid.

Author Response

The manuscript by Scott et al. investigated the relationship between isoflavonoids and free amino acids in soybeans with their resistance to aphids. They tested the concentration of isoflavonoids and free amino acids in 13 aphid resistant, tolerant and suscpetible soybean cultivars. They found that there was a low correlation between isoflavonoid and aphid resistance, but low level of free amino acids might increase the aphid resistance in soybeans. The data in this MS were well described, but there are still issues that need to be addressed. Below please find my detailed comments.

Thank you for the comments and suggestions to improve the manuscript. The responses are in red text.

  • What’s the meaning of “low”, “medium”, “high” and “resistant” in Figure 1. Based on the description, “low”, “medium”, “high” and “resistant” seem to mean the concentration of isoflavonoids. But I am very confusing about the standard of “low”, “medium”, “high” and “resistant”. The concentration of isoflavonoids in RCAT1101a is the highest among the 13 soybean cultivars, but RCAT1101a belongs to “medium”. How to understand here?

The low, medium and high in the X axis label for Figure 1 refer to the categories of isoflavonoid concentration in the leaves of 18 soybean varieties measured at the V1 and V3 stages. This was included in lines 143-145 of the methods. Figure 1 results provide the total isoflavonoid concentrations in plants measured 10 days later where relative changes occurred depending on the variety and aphid pressure. In this case, RCAT 1101a leaf isoflavonoid levels increased compared to other soybean varieties and it became the highest. The E16 265, E16 266 and E16 267 lines were re-labelled as “Aphid Resistant” in the X axis of Figures 1 and 2.

  • Some data in Figure 4 and 5 are not consistent to each other. For example, the aphid counts on top 3 trifoliate leaves of AC Glengarry was low (Figure 4), but the percent damage by soybean aphids to the top 3 trifoliate leaves of AC Glengarry was the highest. How to understand here?

The aphid feeding damage was greater on the AC Glengarry leaves despite the lower aphid numbers because that soybean variety was less tolerant (more susceptible to feeding damage) than other varieties (for example Pagoda, which had greater aphid numbers but less feeding damage).

  • Line 14-15 and Line 33-36: Based on the data, I think this conclusion is not solid.

The soybean varieties determined to be resistant to aphids (fewer aphids counted) were found to not have a strong association with higher isoflavonoid concentrations, but these 2 varieties did have lower amino acid concentrations relative to the more tolerant and susceptible varieties. The authors agree that the 2 statements in the Summary and abstract were confusing, so these are revised: “There was a low correlation between isoflavonoid leaf concentrations and aphid resistance in the soybean varieties studied,  however aphid resistant cultivars were determined to have lower free amino acid concentrations indicating that lower nutrient quality may be responsible for the resistance observed.” (Lines 14-17); “The most susceptible cultivar was Maple Arrow, whereas Pagoda and Conrad were more tolerant to aphid feeding damage. Overall, there was a low correlation between the number of aphids per leaf, feeding damage and leaf isoflavonoid levels.” (Lines 30-34).

Reviewer 3 Report

The article entitled, “investigating soybean genetic resistance associated with anti-soybean aphid activity” by Scott et al. presents interesting data highlighting plant insect interactions and the mechanisms of resistance versus tolerance to soybean aphid feeding.  The authors found little in terms of secondary plant compounds but instead found lower nutritional quality in the resistant cultivars.  These results suggest antibiosis.  The paper is difficult to read and the authors are missing various sets of data that make a complete picture interpretation impossible.  I suggest major revision is needed for this paper

Overall, the results are interesting and are acceptable for publication as they contribute to our understanding of plant insect interactions.  The overall manuscript is well-written, but the authors can improve it through editing to improve readability.  Throughout, the authors use awkward sentence constructs and as written, the sentences do not make sense.  Two examples from the abstract:

Line 32- aphid leaf numbers should be “number of aphids per leaf”.  Aphid leaf number does not exist as a measure.

Line 37-38: Replace:  The findings provide a better understanding of soybean host-plant resistance as well as direction for breeding or metabolic engineering of leaf metabolites that could improve aphid resistance.  It is resistance to aphids not aphid resistance

Suggestions:  The findings provide a better understanding of soybean host-plant resistance and suggest ways to improve soybean resistance to aphid feeding through breeding or metabolic engineering of leaf metabolites.

I suggest that the entire manuscript be carefully edited to improve readability and ensure correctness.

Further improvements to the manuscript:

Title: the title is difficult to sort through.  Maybe something along the lines of, “Investigation of genetic resistance to soybean aphid feeding in soybean cultivars”. 

Methods

Were 5 aphids place on each of the 2nd and 3rd true leaves, or were 5 aphids split between the leaves? Line 144.  How were the controls (lines 149-153) used for the trial assessing aphid survival, growth, and fecundity?  This would be for rating plant response to aphids, but since this was not done, controls for aphids are not needed here. Control plants for isoflavanoids are needed… but is it appropriate to have 4 aphid trials and one plant without aphids and one plant with pipecleaners and paper clips as a control (N =1 control for each type/ combined to two control plants to measure variation)?  You have no estimate of variation in your control plants without aphids…. which could be a serious flaw.  This is especially problematic as you make comparisons of leaf response to aphid feeding.  This needs to be discussed and the decision to use controls in this way must be justified.

Line 161-162. Adult wingless aphids- how old? In 10 day trial, nymphs were used. Why were unknown aged adults used for 4-week trial?

Line 240-243. If data were not able to be normalized, a Kruskal-Wallis test does not allow two-way or three-way test.  KW tests are only one-way.  The appropriate statistic would be Friedman 2-way non-parametric ANOVA.

Why were aphids maintained on “Beer friend” listed as a susceptible in methods (line 129-130) but shown as medium for isoflavanoids (Fig. 1) and shown to not maintain aphids (Fig. 2).  The results from Figure argue selection against soybean aphid (versus using a susceptible like Conrad). 

Add details on percent leaf damage assessment for aphid feeding. Chlorosis? Honeydew production?  How was healthy and damaged leaf area calculated/ determined?

Results

Figure 1. The greens and yellows are difficult to see.  Outside of bars should be lined in black at a minimum.  Why is top leaf missing for Maple Arrow?  The x-axis label is not clear. Low, medium, high, and resistant soybean cultivars.  Isoflavanoid levels- what are they in the resistant cultivars? 

Figure 3. Why 11 cultivars not 13 reported? 

Figure 5, images only allow assessment of 8 cultivars.  The differences in data reported for the different cultivars make interpretation more difficult. Authors have 8 cultivars will all data… consider removing the 5 cultivars that have missing data.

Table 1.  Line 361, authors present “late stage aphid numbers”.  Not quantified or presented earlier.  This is part of the argument for plant rating, but is not quantified or discussed previously.  Leaf damage in same table ranges from low to moderate but figure 4 shows highest level on trifoliate 1 of AC (~30%) , 15% on leaflet 2 and 4% on leaflet 3.  Maple Arrow data are not presented for leaf damage and AC Glengarry is not presented in the table.  Moderate for Harosoy and Avatar have an averae of about 4% and 6.5% but both rate moderate?  This table does not make sense as it is currently presented.

Table 2 presents 5 varieties (out of 13 tested). The table is confusing to compare metabolites across varieties wand with or without aphids.  Same letters mean no difference but it is hard to interpet.  For example.  Max is mean 48, minimum is mean 1.6 but only two are different. Aphids increased the amount across the board.  Shouldn’t authors compare the effect of aphid feeding within a cultivar instead of comparing all cultivars all conditions?  With a T-test, 48 with aphids should be different than 9 without (and more biologically meaningful). 

Is Diadzin the same as dainzein? The spellings differ in table and discussion.

Discussion

Substantial editing and streamlining of discussion is required.  Line 484-488 for example:

Another finding from the present work was that regardless of whether isoflavonoids are relevant to aphid resistance in the selected cultivars, in few cases were leaf concentrations induced due to aphid feeding. The decrease in isoflavonoids by aphid feeding was 486 observed on the 3rd or top leaves within the 10 day trials (Fig. 1), while only AC Colombe and OAC Avatar in the lower isoflavonoid category had significantly increased isoflavo noids after 4 weeks in the aphid-infested leaves (Fig. 3).

Two thoughts in a complex sentence that is hard to follow. Then the next sentence does not relate well with the previous. 

References

A number of capitalizations (Hemiptera, United States, etc. Aphis glycines) require editing.

Author Response

The article entitled, “investigating soybean genetic resistance associated with anti-soybean aphid activity” by Scott et al. presents interesting data highlighting plant insect interactions and the mechanisms of resistance versus tolerance to soybean aphid feeding.  The authors found little in terms of secondary plant compounds but instead found lower nutritional quality in the resistant cultivars.  These results suggest antibiosis.  The paper is difficult to read and the authors are missing various sets of data that make a complete picture interpretation impossible.  I suggest major revision is needed for this paper

Thank you for the many suggestions to improve the manuscript. The responses are in red text and in the manuscript revisions.

Overall, the results are interesting and are acceptable for publication as they contribute to our understanding of plant insect interactions.  The overall manuscript is well-written, but the authors can improve it through editing to improve readability.  Throughout, the authors use awkward sentence constructs and as written, the sentences do not make sense.  Two examples from the abstract:

  • Line 32- aphid leaf numbers should be “number of aphids per leaf”.  Aphid leaf number does not exist as a measure.

This was revised in the text (Line 33).

  • Line 37-38: Replace:  The findings provide a better understanding of soybean host-plant resistance as well as direction for breeding or metabolic engineering of leaf metabolites that could improve aphid resistance.  It is resistance to aphids not aphid resistance
  • Suggestions:  The findings provide a better understanding of soybean host-plant resistance and suggest ways to improve soybean resistance to aphid feeding through breeding or metabolic engineering of leaf metabolites.

The sentence was revised as per the suggestions to read  “The findings provide a better understanding of soybean host-plant resistance and suggest ways to improve soybean resistance to aphid feeding through breeding or metabolic engineering of leaf metabolites.” (Lines 38-41)

  • I suggest that the entire manuscript be carefully edited to improve readability and ensure correctness.

The authors agree  - The manuscript was revised to improve the readability..

Further improvements to the manuscript were made, for example in the Discussion section where “Differences in leaf damage were observed due to characteristics of antibiosis, antixenosis or both for each cultivar” – was changed to “Upon aphid infestation to soybean cultivars, differences in leaf damage were observed which may be due to characteristics of antibiosis, antixenosis or both for each cultivar”. (Lines 452-454)

Title: the title is difficult to sort through.  Maybe something along the lines of, “Investigation of genetic resistance to soybean aphid feeding in soybean cultivars”. 

The title was revised to read “Investigation of metabolic resistance to soybean aphid (Aphis glycines Matsumura) feeding in soybean cultivars”

Methods

  • Were 5 aphids place on each of the 2nd and 3rd true leaves, or were 5 aphids split between the leaves? Line 144. 

The sentence was edited to read “Five newly emerged aphid nymphs (< 1 day old) were placed on the mid leaf of both the 2nd and 3rd true leaves…” (Lines 146-147)

  • How were the controls (lines 149-153) used for the trial assessing aphid survival, growth, and fecundity?  This would be for rating plant response to aphids, but since this was not done, controls for aphids are not needed here. Control plants for isoflavanoids are needed… but is it appropriate to have 4 aphid trials and one plant without aphids and one plant with pipecleaners and paper clips as a control (N =1 control for each type/ combined to two control plants to measure variation)?  You have no estimate of variation in your control plants without aphids…. which could be a serious flaw.  This is especially problematic as you make comparisons of leaf response to aphid feeding.  This needs to be discussed and the decision to use controls in this way must be justified.

This question was also posed by reviewer 1: The control plants with and without the set-up were single replicates and were pooled together to represent the range of isoflavonoid leaf concentrations under both conditions. The average concentration and standard error for the both controls was then compared to the 4 replicate aphid-infested plants for each cultivar. The comparison of the control to aphid-infested plants allowed for an assessment of whether the aphids induced isoflavonoid levels above the control levels (both types). This point was added in the text (Lines 154-155).

  • Line 161-162. Adult wingless aphids- how old? In 10 day trial, nymphs were used. Why were unknown aged adults used for 4-week trial?

Aphid adults were randomly selected from the colony  - this was revised in the text to read “…two wingless adult aphids, randomly selected from the colony, placed on the first full mid leaf…” (Lines 166-167).

  • Line 240-243. If data were not able to be normalized, a Kruskal-Wallis test does not allow two-way or three-way test.  KW tests are only one-way.  The appropriate statistic would be Friedman 2-way non-parametric ANOVA.

The explanation of the statistics used to analyze differences in the number of aphids on the soybean leaf in the 10 day trial was not correct – the K-W test was replacing a one way ANOVA as the aphids counted were only the ones on the 3rd leaf of each plant. This was revised in the text to read “In the case of the aphid counts on the 3rd leaf of soybean plants in the 10 day trial, a non-parametric Kruskal-Wallis test was used with comparisons performed with a Dunn’s test” (Lines 248-250).  

  • Why were aphids maintained on “Beer friend” listed as a susceptible in methods (line 129-130) but shown as medium for isoflavanoids (Fig. 1) and shown to not maintain aphids (Fig. 2).  The results from Figure argue selection against soybean aphid (versus using a susceptible like Conrad). 

This point was also raised by Reviewer 1:  Soybean cv. Beer Friend was not considered a susceptible control, rather it was included as part of the study since the soybean aphids had been maintained on BF for many generations prior to the beginning of the study. There was no expectation that the aphids would be more or less susceptible to BF, rather they would be adapted to it. This would not necessarily mean they would be more tolerant or resistant to BF compared to other soybean lines that could be more susceptible or resistant.

  • Add details on percent leaf damage assessment for aphid feeding. Chlorosis? Honeydew production?  How was healthy and damaged leaf area calculated/ determined?

No significant amounts of chlorosis or honey dew were observed on the aphid infested leaves. The details were added to the text “For each cultivar, new assessment parameters were established in WinFOLIA for the colour of the healthy control leaves first, and then the colour of aphid damaged areas on the infested leaves. The program analyzed the amount of healthy leaf colour on the aphid damaged leaf which was then converted to percent leaf damage based on the difference” (Lines 183-187).

Results

  • Figure 1. The greens and yellows are difficult to see.  Outside of bars should be lined in black at a minimum.  Why is top leaf missing for Maple Arrow?  The x-axis label is not clear. Low, medium, high, and resistant soybean cultivars.  Isoflavanoid levels- what are they in the resistant cultivars? 

One of the green bars was lined in black to improve the contrast. The low, medium and high in the X axis label for Figure 1 refer to the categories of isoflavonoid concentration in the leaves of 18 soybean varieties measured at the V1 and V3 stages. This was included in lines 143-145 of the methods and in the Figure 1 legend. The isoflavonoid levels for the 3 aphid-resistant soybean cultivars were added separately as the V3 analysis was not completed as it had been for the other 13 cultivars. The comparison was completed only after the 10 day trial with the other 13 cultivars.

  • Figure 3. Why 11 cultivars not 13 reported? 

Reviewer 1 also asked this question: The 4 varieties chosen for each of the 3 greenhouse trials were selected based on the isoflavonoid leaf levels measured at the V1 stage in the original 18 cultivars that were screened. At least one representative cv. from each low, moderately high and high isoflavonoid grouping was included within each trial. Of the 12 cultivars initially included in the 3 trials, one variety (OAC Strive – considered to have moderately high isoflavonoid levels) did not grow well up to the  4th week and was thus dropped from study.  This point was added to the text (Lines 300-301).

  • Figure 5, images only allow assessment of 8 cultivars.  The differences in data reported for the different cultivars make interpretation more difficult. Authors have 8 cultivars will all data… consider removing the 5 cultivars that have missing data.

Unfortunately the leaf damage results for 3 of the cultivars was lost – only 8 of the original 12 cultivars tested in the 4 week greenhouse trial could be compared in this case, Although the inclusion of the same 8 cultivars in Figures 4 and 5 would allow for a more straightforward interpretation the authors strongly believe it is important to include as many of the original cultivars to increase the power of the statistical analyses.

  • Table 1.  Line 361, authors present “late stage aphid numbers”.  Not quantified or presented earlier.  This is part of the argument for plant rating, but is not quantified or discussed previously.  Leaf damage in same table ranges from low to moderate but figure 4 shows highest level on trifoliate 1 of AC (~30%) , 15% on leaflet 2 and 4% on leaflet 3.  Maple Arrow data are not presented for leaf damage and AC Glengarry is not presented in the table.  Moderate for Harosoy and Avatar have an averae of about 4% and 6.5% but both rate moderate?  This table does not make sense as it is currently presented.

Table 1 is not meant to provide a quantitative comparison of the 5 cultivars, rather a qualitative one based on the information available from the early growth stage (10 day trial) and later stage (4 week trial) aphid counts and leaf feeding damage to select cultivars representative of the 3 categories of resistance examined. The table and text (Lines 359-361) were revised to provide a better description of each cultivar.

  • Table 2 presents 5 varieties (out of 13 tested). The table is confusing to compare metabolites across varieties wand with or without aphids.  Same letters mean no difference but it is hard to interpet.  For example.  Max is mean 48, minimum is mean 1.6 but only two are different. Aphids increased the amount across the board.  Shouldn’t authors compare the effect of aphid feeding within a cultivar instead of comparing all cultivars all conditions?  With a T-test, 48 with aphids should be different than 9 without (and more biologically meaningful). 

The purpose of Table 2 was to provide all of the metabolite data in order to compare differences across the 5 cultivars with and without aphid feeding. The significant findings were then provided in the results section. The authors are not certain which FAAs the reviewer is referring to in the example, in most cases where there was no difference in FAA levels with and without aphids, the large variability between the 3 replicates was likely responsible, perhaps with more replication the effect of aphid feeding would have been more significant.

  • Is Diadzin the same as dainzein? The spellings differ in table and discussion.

Daidzin is the glycoside of the aglycone, daidzein, The spelling in the text was correct, but the spelling in Table 2 was corrected.

Discussion

  • Substantial editing and streamlining of discussion is required.  Line 484-488 for example:
  • Another finding from the present work was that regardless of whether isoflavonoids are relevant to aphid resistance in the selected cultivars, in few cases were leaf concentrations induced due to aphid feeding. The decrease in isoflavonoids by aphid feeding was 486 observed on the 3rd or top leaves within the 10 day trials (Fig. 1), while only AC Colombe and OAC Avatar in the lower isoflavonoid category had significantly increased isoflavo noids after 4 weeks in the aphid-infested leaves (Fig. 3).

The two sentences were revised in the text to read “Another finding from the present work was that aphid feeding had little effect on isoflavonoid levels, regardless of whether it was a cultivar considered to be resistant or susceptible to aphids. In only 2 cases did aphid feeding significantly alter isoflavonoid levels: AC Colombe and OAC Avatar, both in the lower isoflavonoid category, had significantly higher isoflavonoids on aphid infested leaves compared to controls after 4 weeks (Fig. 3)” (Lines 508-516). 

  • Two thoughts in a complex sentence that is hard to follow. Then the next sentence does not relate well with the previous. 

References

  • A number of capitalizations (Hemiptera, United States, etc. Aphis glycines) require editing.

 Corrections to capitalizations were made in the References.

Round 2

Reviewer 2 Report

The authors followed my advice and answer my questions in the revised version.